Effect of mesoporous bioactive glass on odontogenic differentiation of human dental pulp stem cells

Zhu Lin
Li Jingyi
Dong Yanmei kqdongyanmei@bjmu.edu.cn
Department of Cariology and Endodontology, Peking University School and Hospital of Stomatology & National Clinical Research Center for Oral Diseases & National Engineering Laboratory for Digital and Material Technology of Stomatology & Beijing Key Laboratory of Digital Stomatology , Beijing , China
Leppik Liudmila
Electronic publication date: 2021 Nov 23
Publication date: 2021
Volume: 9
Electronic Location ID: e12421
Received 2021 Jul 12; Accepted 2021 Oct 11
Copyright: © 2021 Zhu et al.
Copyright year: 2021
Copyright holder: Zhu et al.
License: This is an open access article distributed under the terms of the Creative Commons Attribution License, which permits unrestricted use, distribution, reproduction and adaptation in any medium and for any purpose provided that it is properly attributed. For attribution, the original author(s), title, publication source (PeerJ) and either DOI or URL of the article must be cited.
License URL: https://creativecommons.org/licenses/by/4.0/

Keywords: Mesoporous bioactive glass, Pulp cells, Dental pulp dentin complex

Funding: National Natural Science Foundation of China 51372005 This work was supported by the National Natural Science Foundation of China (51372005). The funders had no role in study design, data collection and analysis, decision to publish, or preparation of the manuscript.

==============================
Healthy pulp tissue plays an important role in normal function and long-term retention of teeth. Mesoporous bioactive glass (MBG) as a kind of regenerative biomaterials shows the potential in preserving the vital pulp. In this study, MBG prepared by organic template method combined with sol-gel method were used in human dental pulp cell culture and ectopic mineralization experiment. Real-Time PCR was used to explore its ability to induce odontogenic differentiation of dental pulp cells. MBG and rat crowns were implanted under the skin of nude mice for 4 weeks to observe the formation of pulp dentin complex. We found that MBG can release Si and Ca ions and has a strong mineralization activity in vitro. The co-culture of MBG with human dental pulp cells promoted the expression of DMP-1 (dentin matrix protein-1) and ALP (alkalinephosphatase) without affecting cell proliferation. After 4 weeks of subcutaneous implantation in nude mice, the formation of hard tissue with regular structure under the material could be seen, and the structure was similar to dentin tubules. These results indicate that MBG can induce the differentiation of dental pulp cells and the formation of dental pulp-dentin complex and has the potential to promote the repair and regeneration of dental pulp injuries.

Introduction

Healthy pulp tissue plays an important role in normal function and long-term retention of teeth. The principle of treatment of periapical diseases of pulp should preserve the vital pulp as much as possible and restore the structure and function of the dental pulp-dentin complex. Bioactive glass (BG) is a kind of inorganic material with silicon, calcium and phosphorus as the basic system. Previous studies have confirmed that BG has a good application prospect in the repair and regeneration of skin and bone tissue injuries (Lalzawmliana et al., 2020; Kargozar et al., 2020).

Our previous studies have shown that a serials of micro-nano sized BG prepared by sol-gel method can promote migration, proliferation, odontogenic differentiation and mineralization of human dental pulp cells (Mi, Dong & Gao, 2012). The rat molars were directly covered with the pulp capping agent containing micro-nano BG, and the formation of a restorative dentine bridge with the dentin tubulo-like structure at the exposed pulp hole was observed at 4 weeks (Long et al., 2017; Cui et al., 2017). These BG have uneven particle size distribution, poor dispersion and agglomeration, which to a certain extent limits the performance and clinical application effect of BG. In order to improve the morphology of micro-nano BG and increase its biological activity, the particles can be prepared into porous materials by using sol-gel method combined with specific organic template technology (Hu et al., 2014a). These porous BG particles with more uniform particle size, smaller pore size and better dispersion can be divided into mesoporous materials. The mesoporous structure in mesoporous bioactive glass (MBG) increases the specific surface area of the material so much that the ions exchange rapidly after contact with the liquid, which is conducive to the release of the bioactive ions and can be decomposed more easily (Fiume et al., 2018). Yan et al. (2004) found that hydroxyapatite could be formed after MBG particles were soaked in SBF solution for 4 h, which has a stronger ability to promote mineralization in vitro. Meanwhile, MBG can also be used as a carrier to load drugs or growth factors to form a local microenvironment that is conducive to the proliferation and differentiation of dental pulp cells. In addition, MBG can also promote the proliferation of osteoblasts/fibroblasts, promote the osteogenic differentiation of bone marrow mesenchymal stem cells, and enhance the activity of alkaline phosphatase (Lalzawmliana et al., 2020; Hu et al., 2018). However, the effect of MBG on the regeneration of dental pulp dentine complex is still rarely confirmed by relevant studies.

In this study, MBG prepared by organic template method combined with sol-gel method were used in human dental pulp cell culture and ectopic mineralization experiment, to explore its ability to induce odontogenic differentiation of dental pulp cells, and to explore its application prospect in the field of dental pulp injury repair and regeneration.

Methods

Preparation and characterization of MBG

MBG (molar percentage of chemical composition is 60% SiO2, 36% CaO and 4% P2O5) was prepared by sol-gel method combined with organic template technology. The preparation method was briefly described as follows. Hexadecyl trimethyl ammonium bromide (CTAB; Guangzhou Chemical Reagent Factory, Guangdong, China) was added to a mixture of deionized water and ethanol to form CTAB micelles, and then tetraethyl silicate (TEOS; Guangzhou Chemical Reagent Factory, Guangdong, China) dissolved in cyclohexane was added. Adding ammonia water initiates the reaction to promote TEOS hydrolysis. The mixture was stirred, and triethyl phosphate (TEP; Guangzhou Guanghua Chemical Reagent Co. LTD, Guangdong, China) and calcium nitrate tetrahydrate (CN; Guangzhou Chemical Reagent Factory, Guangdong, China) were added sequentially to form a white precipitate. The precipitate collected by filtration was washed with ethanol and deionized water, and dried at room temperature. Finally, MBG is obtained after removal of organic matter and nitrate by calcination at 650 °C. The material was prepared by National Engineering Research Center for Tissue Restoration and Reconstruction, South China University of Technology.

Field emission scanning electron microscopy (FE-SEM, S4800, JEOL, Japan) was used to observe the surface structure of MBG. The samples were sprayed with platinum at 50 mTorr for 5 min before being observed by FE-SEM. Transmission Electron Microscope (TEM; JEM-2100HR, JEOL, Japan) was used to observe the pore morphology and internal structure of MBG. In the previous study (nitrogen desorption experiment), the specific surface area of MBG particles was 540.400 m2/g, and the average pore diameter was about 7.325 nm (Zhu et al., 2018).

Physicochemical properties of MBG

MBG and 45S5 bioglass were used as the experimental group and the control group for the study of physicochemical properties. 45S5 is a traditional bioactive glass and has been widely used in clinical practice, but it still has some limitations such as strong alkalinity that may cause tissue necrosis, so it is used as a control group to evaluate whether the newly prepared MBG has certain improve.

MBG particles and 45S5 bioglass (molar percentage of chemical composition is 45% SiO2, 24.5% CaO, 6% P2O5 and 24.5% NaO) particles were placed in a 180 °C constant temperature drying oven for sterilization at high temperature for 4 h. They were added to dulbecco’s modified eagle medium (DMEM; Gibco; Thermo Fisher Scientific, Waltham, MA, USA; pH = 7.40) at concentrations of 0.1 and 1 mg/mL respectively and shaken at 37 °C and 120 rpm to obtain homogeneous suspension. At 1, 3, 6, 12 and 24 h, pH values of the suspension in each group were detected with a pH meter (PH3-3C, yueping Inc, shanghai, China), and pH curves of each group were drawn. The experiment was repeated for three times independently.

The sterilized MBG particles and 45S5 BG particles were added into the DMEM at a concentration of 0.1 mg/mL. After being swished at 37 °C at 120 rpm for 24 h, the supernatant was centrifuged at 14,000 g/min for 5 min. After the supernatant was filtered with a 0.22 m filter, five mL was taken and the concentration of Si, Ca and P ions in each group was tested by inductively coupled plasma analysis (ICP, ICAP 6300, Thermo, German).

The ability of MBG to form hydroxyapatite in vitro was determined by FE-SEM, Fourier Transform Infrared Spectroscopy (FTIR; Nexus, Washington, DC, USA) and X-Ray Diffraction (XRD, XPert Pro MPD, Panaco, Netherlands). MBG particles were added into the solution of simulated body fluid (SBF) at a concentration of 1 mg/mL (the ion concentrations were Na+ 142 mmol/L, K+ 5.0 mmol/L, Ca2+ 2.5 mmol/L, Mg2+ 1.5 mmol/L, Cl− 147.8 mmol/L, HCO3− 1.0 mmol/L, SO42− 0.5 mmol/L, pH = 7.25–7.45). After shaking at 37 °C and 120 rpm for 24 h, the supernatant was discarded by centrifugation at 3,500 g/min for 5 min. The remaining solid MBG powder was removed. After washing and precipitation with acetone and deionized water for three times, the powder was dried at 37 °C in a vacuum drying oven. The mineral deposition on the surface was observed by FE-SEM. Infrared absorption spectra were measured by FTIR. The XRD pattern was obtained by Xpertpro X-ray diffraction analyzer. Cu target Kα ray was used to scan the sample from 10° to 80°, tube voltage was 40 kV, tube current was 100 mA. Hydroxyapatite (HA) standard card was used as control to analyze the mineral composition.

Effects of MBG on proliferation, differentiation of dental pulp cells

MBG particles were sterilized by dry heat in 180 °C incubator for 4 h, then cooled to room temperature for later use. The homogeneous suspension was prepared by adding MBG into anhydrous ethanol at 0.1 mg/mL and shaken at 37 °C and 120 rpm. The MBG suspension was added into 96-well plate and 12-well plate at the volume of 100 μL/well and 1,400 μL/well, respectively. In the control group, equal volume of anhydrous ethanol was added to the culture plates, and the plates were placed in a UV ultra clean table to be air-dried overnight to ensure that the BG particles was firmly attached to the bottom of the culture plates. The plates were sterilized by ultraviolet radiation for 30 min for later use.

Human dental pulp cells (hDPCs) were isolated from freshly extracted impacted wisdom teeth of patients with informed consent from the Department of Oral and Maxillofacial Surgery as previously described (PKUSSRB-202053006) (Gronthos et al., 2000). Cells were cultured in DMEM containing 10% fetal calf serum (FBS; Kangyuan, Tianjin, China), 1% penicillin and streptomycin (Gibco) in the incubator at 37 °C and 5% CO2. A total of 0.25% trypsine-EDTA solution was used for digestion when the cell reached 80%. The cell culture medium was changed every 2 days. Cells in passages 4–6 were used for all the experiments.

The proliferation of hDPCS was detected by MTT assay. The cells were seeded at the density of 3 × 103 cells/well in 96-well plates (Corning, Corning, NY, USA) pretreated with MBG suspension or anhydrous ethanol, with five wells per group. The cells were incubated at 37 °C and 5% CO2, and the liquid was changed every other day. On days 1, 3, 5, and 7, 180 μL fresh DMEM and 20 μL MTT (Sigma, St. Louis, MO, USA) solution at a concentration of five mg/mL were added to each well for 4 h at 37 °C and 5% CO2. After that, 150 μL DMSO (Sigma) was added to each well and the plates were shaken at 37 °C for at least 10 min until the crystals were dissolved. The optical density (OD) was measured at 490 nm by a microplate analyzer.

HDPCs were seeded in 12-well plates pretreated with anhydrous ethanol or MBG suspension at a density of 1 × 105 per well and cultured at 37 °C and 5% CO2. After 7 days, Total RNA was extracted using Trizol (TRIzol, Invitrogen, Waltham, MA, USA) following manufacturer’s instructions. Using reverse transcription kit, cDNA was synthesized by the Prime Script RT Master Mix (Takara, Tokyo, Japan). Target genes were dentin matrix protein (DMP-1) and alkaline phosphatase (ALP), glyceraldehyde-phosphate dehydrogenase (GAPDH) was used as internal reference, and primer sequences were shown in Table 1. A total of 20 μL of amplification reaction system was pre-denaturated and activated at 95 °C for 3 min, denaturated at 95 °C for 3 s, and annealed and extended at 60 °C for 20 s for 40 cycles in total in the ABI QuantStudio three Real-Time PCR.

Table 1 Primer sequences.

Gene	Sequence(5′-3′)	
ALP	Forward: AGCACTCCCACTTCATCTGGAA
Reverse: GAGACCCAATAGGTAGTCCACATTG	
DMP-1	Forward: AGGAAGTCTCGCATCTCAGAG
Reverse: TGGAGTTGCTGTTTTCTGTAGAG	
GAPDH	Forward: GAAGGTGAAGGTCGGAGTC
Reverse: GAGATGGTGATGGGATTTC	

MBG induced the formation of dental pulp dentine complex in rats

The research scheme passed the ethical review of the Biomedical Ethics Committee of Peking University (LA2011-057). All the experimental animals were obtained from the Beijing Vital River Laboratory Animal Technology Co., Ltd. The experimental process and the animal’s feeding, nursing and living conditions are all in the SBF environment. Four male nude mice in one cage were included in this observational experiment.

The 4-week-old SD rats were euthanized by overdose of sodium pentobarbital. The head of the rats was soaked in 75% alcohol for 10 s, the upper and lower jaws were surgically separated, and sterile PBS was washed for three times. The upper and lower first molars were separated from the jaws and rinsed with sterile PBS for three times. The root of the molar was removed from the neck of the tooth with a surgical blade, and the crown and pulp tissue were preserved. The molar was placed in fresh DMEM medium for use. The sterilized MBG particles (two mg) were covered in the crown pulp section as the experimental group, and the crown implant alone as the control group. Four samples were set in each group.

Male BALB/c nude mice aged 6 weeks were anesthetized by intraperitoneal injection of 0.5 % pentobarbital sodium 30 mg/kg. The surgical area was disinfected with 75% alcohol, and three mm transverse incisions were made on both sides of the back near the hind legs. The hemostatic forceps were blunt to separate the subcutaneous tissue and form pouches. A random sample was implanted in each pouch. The incisions were sutured, and the nude mice were put back into SPF environment for further feeding after they regained consciousness. After 4 weeks of experiment, all nude mice were killed by inhaling excessive carbon dioxide, the samples were taken out, dehydrated, paraffin-embedded, and sliced along the long axis of the tooth with a thickness of five μm. The spread pieces, patches and patches were baked at 58 °C for 2 h and then observed by HE staining (Baso Diagnostic, Inc., Zhuhai, Guangdong, China).

Statistical analysis

SPSS 24.0 was used for statistical analysis. Overall analysis of pH value, cell proliferation and gene expression in each group were analyzed by One-way ANOVA. LSD test was used for comparison between groups, and P < 0.05 was considered statistically significant.

Results

Morphology of MBG

MBG particles were observed by FE-SEM, as shown in Fig. 1A. MBG was a spherical particle with regular morphology, rough surface and good dispersion. The particle size was sub-micron and the size was about 300–500 nm. The mesoporous structure inside MBG particles was observed by TEM, as shown in Fig. 1B. Regularly and uniformly distributed radial mesoporous structure can be seen inside MBG particles.

Figure 1 The morphology and physicochemical properties of 60S MBG.

(A) Particle scanning electron microscopy of MBG; (B) transmission electron microscopy of MBG; (C) the extraction pH curve of MBG; (D) scanning electron microscopy after MBG immersion of simulated body fluids; (E) the surface sediment infrared spectrum of MBG; (F) the surface sediment X-ray diffraction pattern of MBG.

Physical and chemical properties of MBG

The pH change curve of the extracted solution with different concentrations of MBG and 45S5 for 24 h was detected by pH meter, and the results were shown in Fig. 1C. The pH of MBG extract increased with the increase of extraction time and reached the platform stage 6 h later. When the concentration was 0.1 mg/mL, the pH value of MBG extract increased from 7.40 to 7.70 in the first 1 h, and then maintained at about 7.90 at the platform stage. The pH value of the 45S5 extract increased from 7.40 to 7.84 in the first 1 h, then reached the platform stage and maintained at about 8.10. The pH of 45S5 group was significantly higher than that of MBG group at the same extraction time point (P < 0.05).

After 0.1 mg/mL MBG was extracted for 24 h, the concentration of Si, Ca and P ions in the solution was detected three times by ICP and the mean and standard deviation were provided (Table 2). As shown in Table 2, MBG could significantly improve the concentration of Si ions in the solution.

Table 2 Ion concentration of MBG extract.

	Si (ppm)	Ca (ppm)	P (ppm)	
DMEM (BLK)	0.076 ± 0.002	66.64 ± 1.90	27.61 ± 0.77	
45S5	12.14 ± 1.29	72.59 ± 0.49	27.48 ± 0.23	
MBG	31.81 ± 0.68	51.54 ± 2.45	19.41 ± 1.22	

FE-SEM was used to observe the mineral deposition on the surface of MBG, as shown in Fig. 1D. After soaking in SBF for 24 h, a large number of short-stick mineral deposits could be seen on the surface of MBG particles. FTIR was used to detect and analyze the mineral components deposited on MBG surface, as shown in Fig. 1E. The results showed that after MBG was immersed in SBF for 24 h, a dispersion peak at 580 cm−1 on the FTIR was split into two peaks, which were located at 603 and 562 cm−1 respectively, representing a typical bimodal structure formed by the bending vibration of crystallized P-O bond. Hint: After MBG was soaked in SBF for 24 h, the surface began to deposit minerals containing hydroxyapatite crystals. Fig. 1F shows the XRD pattern of MBG soaked in SBF for 24 h. After soaking for 1 day, the sample showed a diffraction peak at 2 at 26° and 32°. According to the standard PDF card JCPDS-09-0432, the diffraction peak was hydroxyapatite (HA) diffraction peak. A total of 26° and 32° correspond to (002) and (211) crystal surfaces respectively. This indicates that HA is formed on the surface of MBG.

Effects of MBG on hDPCs proliferation and differentiation

MTT assay was used to detect the number of living cells cultured with 0.1 mg/mL MBG at different time points, and the results showed that there was no statistical difference between the cell proliferation curve after MBG treatment and the control group. It indicates that MBG 0.1 mg/mL does not affect hDPCs proliferation (Fig. 2A).

Figure 2 The effect of MBG on odontogenic differentiation of hDPCs.

(A) The effect of MBG on hDPCs proliferation detected by MTT; (B) the effect of MBG on hDPCs odontogenic differentiation genes ALP and DMP-1 detected by RT-PCR. *P < 0.05.

The expressions of ALP and DMP-1 of hDPCs dentin differentiation genes were detected by Real-Time PCR, and the results showed that the expressions of ALP and DMP-1 in MBG group were significantly higher than those in the control group at 7 days (P < 0.05). This indicated that MBG culture promoted the differentiation of hDPCs (Fig. 2B).

MBG induced the formation of endodontic dentin complex in rats

A total of 4 weeks after in vivo transplantation, HE staining was shown in Fig. 3. In the control group, there was mild inflammatory infiltration in pulp, and the number of pulp cells was small. The pulp section has a reddish fibre-like matrix with no visible hard tissue formation. In the pulp section of the MBG group in contact with the material, obvious hard tissue formation and regular structure can be seen, and a small amount of cavitation formed after MBG demineralization.

Figure 3 The formation of endodentine complex induced by MBG in rats.

The formation of endodentine complex induced by MBG in rats (Note: the figure below is enlarged in the solid frame of the figure above. Abbreviation: D, dentin; P, pulp tissue; RD, reparative dentine, DL, dentin tubule-like structure, OL, osteoid dentin structure).

There was also a homogeneous layer of secondary dentine deposition near the pulp cavity. At high magnification, it can be seen that the mineralized matrix in the pulp section has dentin tubule-like structure. A few cells’ cytoplasm protrude into the mineralized matrix to form the complex structure of pulp dentin.

Discussion

In this study, the surfactant CTAB was used as the template of mesoporous structure to prepare MBG with loose and divergent structure in the emulsion reaction system. It is characterized by uniform size, good dispersion and good in vitro mineralization induced activity. During the preparation of MBG, glassy sol particles were introduced into the CTAB induced system to polymerize inside their micelles. After volatilization of CTAB, a pore structure is formed inside the glass particles (Hu et al., 2014b). Traditional silicate glass has a complete network structure formed by -Si-O-Si- bridge oxygen bonds, which is relatively stable (Serra et al., 2002). The incorporated Ca is introduced into the system to interrupt the covalent network structure, changing the -Si-O-Si- bridge oxygen bond into a non-bridging oxygen bond -Si-O-M+ (M+ is the modified cation), forming a partially open grid structure (Brauer, 2015; Tilocca & Cormack, 2010). The combination of the internal components becomes loose, the template agent is easy to enter the interior, the connection between the small mesopores is opened, and the pore size is increased. In contact with liquid, there are more cations such as Ca2+ in contact with non-bridging oxygen and body fluids for rapid ion exchange, accelerating the formation of hydroxyapatite, thus showing better biocompatibility and bioactivity.

The average pore diameter of MBG is about 7.325 nm. Kumar found that MBG with pore sizes of few nano meters exhibit favorable biocompatibility in vitro behavior and found to be promising candidate in the field of biomaterials including tissue regeneration and drug storage (Kumar, Aditya & Murugavel, 2019). MBG has a relatively stable pH. The cations released by BG can be replaced by H+/H3O+ in aqueous solution through ion exchange to increase pH (Hench, 2006). The Ca2+ in MBG 24-h extract is significantly lower than 45S5, and the effect of increasing pH at a relatively high concentration (1 mg/mL) is also significantly lower than 45S5. It was found that the pH value of 6.6 to 7.8 was more suitable for the growth of pulp cells. pH beyond this range can cause inflammatory responses in cells, leading to the death of pulp cells (Hirose et al., 2016). In this study, the effect of MBG on pH increase was significantly weaker than that of 45S5, suggesting that MBG has better biocompatibility with tissues and less influence on the growth of pulp cells in vivo.

MBG can release a higher level of Si, induce the expression of odontogenic genes in pulp cells and form a pulp dentin complex. A higher level of Si can promote the proliferation of dental pulp cells, promote the expression of cell surface integrin 2 and 1 (Liu et al., 2014), promote the secretion of extracellular matrix of dental pulp, activate relevant downstream signaling pathways, and promote the differentiation of dental pulp cells into teeth (Wu et al., 2014). The relatively stable weakly alkaline environment provided by MBG also contributes to the odontogenic differentiation of pulp cells. Similar results (Wang et al., 2014; Ravanbakhsh et al., 2019) showed that MBG has good tissue induction, and has a faster degradation rate compared with the traditional 45S5 and 58S BG, which is basically matched with the tissue repair rate.

This study confirmed that MBG can promote the differentiation and mineralization of dental pulp cells through in vitro and nude mouse subcutaneous experiments. At the same time, due to its relatively large specific surface area, mesoporous materials may have a faster degradation rate after contact with body fluids. Therefore, animal experiments may require a shorter observation time to observe the early effects and degradation of the materials on dental pulp cells. On the other hand, the large specific surface area of MBG demonstrates its excellent potential as a drug carrier material. The MBG compound drug may be made into a bioactive pulp capping agent with anti-inflammatory and antibacterial effects for more extensive application in vital pulp preservation.

Conclusions

MBG prepared by using CTAB as a template has relatively regular size and good dispersibility, large specific surface area, and relatively stable pH in solution. It can provide a suitable environment for the differentiation of human dental pulp cells into odontogenesis, and has a certain effect of inducing the formation of dental pulp-dentin complex, and has a good application prospect in the field of dental pulp damage repair and regeneration.

Supplemental Information

Supplemental Information 1 The morphology and physicochemical properties of 60S MBG, the effect of MBG on odontogenic differentiation of hDPCs and the formation of endodentine complex induced by MBG in rats.

Click here for additional data file.

Supplemental Information 2 The in vivo study process of MBG-induced rat pulp-dentin complex formation.

Click here for additional data file.

Supplemental Information 3 ARRIVE Author checklist.

Click here for additional data file.

We also thank Prof. Xiaofeng Chen and Dr. Yudong Wang from National Engineering Research Center for Tissue Restoration and Reconstruction, Southwest University of Science and Technology, Guangzhou 510006, China.

Additional Information and Declarations

Competing Interests

Author Contributions

Animal Ethics

Data Availability

The authors declare that they have no competing interests.

Lin Zhu conceived and designed the experiments, performed the experiments, analyzed the data, prepared figures and/or tables, authored or reviewed drafts of the paper, and approved the final draft.

Jingyi Li conceived and designed the experiments, performed the experiments, analyzed the data, prepared figures and/or tables, and approved the final draft.

Yanmei Dong conceived and designed the experiments, authored or reviewed drafts of the paper, and approved the final draft.

The following information was supplied relating to ethical approvals (i.e., approving body and any reference numbers):

The research scheme passed the ethical review of the Biomedical Ethics Committee of Peking University (LA2011-057).

The following information was supplied regarding data availability:

The raw measurements are available in the Supplementary Files.

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
