# Peer review of "Effect of mesoporous bioactive glass on odontogenic differentiation of human dental pulp stem cells"

_PeerJ, doi:10.7717/peerj.12421_

## Round 0.1 · original submission · Major Revisions

Please provide answers to reviewers' comments and apply necessary changes to the manuscript.

Reviewer 1 ·

Basic reporting

The study investigated the effects of mesoporous bioactive glass (MBG) on odontogenic differentiation of human dental pulp stem cells. This work expands the applications of MBG to dental applications, and the results show the potential of MBG in dental pulp damage repair and regeneration. In my opinion, this paper could be accepted for publication in PeerJ after some issues are addressed. Here below are the specific comments:

Introduction


1. Line 49, the authors stated, “… the particles can be prepared into mesoporous materials with more uniform particle size, smaller pore size, and better dispersion by using sol-gel method combined with organic template technology.” Here the definition of MBG should be clarified. Actually, not all MBG have uniform particle size and better dispersion. Only these MBG produced under certain conditions have a uniform spherical shape and favorable dispersity (Ref. Porous bioactive glass micro- and nanospheres with controlled morphology: developments, properties and emerging biomedical applications, Mater. Horizons. 8 (2021) 300–335.). The authors should clarify this.

Experimental design

Methods

2. Section: 2.2 Physicochemical properties of MBG: The composition of 45S5 Bioglass and their particle size should be given. Also, why did the authors compare 45S5 Bioglass and MBG in terms of their pH and ion release behavior? The authors should clarify it.

3. Line 105 “The remaining solid bioglass powder was removed” here “bioglass” should be “MBG” if I understand correctly.

4. “MBG induced the formation of dental pulp dentine complex in rats”. I suggest that the authors could use a schematic to describe the process of in vivo study to help readers understand the procedure.

Validity of the findings

Results and discussion

5. Table 1, why the concentration of P decreased in MBG group compared to the control?

6. Line 241, “Traditional silicate glass has a complete network structure formed by -Si-O-Si- bridge oxygen bonds, which is relatively stable[13]. CTAB is introduced into the system to interrupt the covalent network structure, changing the -Si-O-Si- bridge oxygen bond into a non-bridging oxygen bond -Si-O-M+ (M+ is the modified cation), forming a partially open grid structure[14, 15].” Here the explanation was wrong. CTAB did not change the -Si-O-Si- bridge oxygen bond into a non-bridging oxygen bond -Si-O-M+ (M+ is the modified cation). The incorporated Ca changed it in your case.

7. Line 258, “ In this study, the effect of MBG on pH increase was significantly weaker than that of 45S5, suggesting that MBG has better biocompatibility with tissues and less influence on the growth of pulp cells in vivo.” It is not clear to me why the authors compared MBG with 45S5 Bioglass? Has 45S5 been used clinically? If in this case, I suggest that the authors should also compare MBG with 45S5 in cell and animal studies.

Reviewer 2 ·

Basic reporting

In this manuscript, the authors reported that MBG could induce the odontogenic differentiation of dental pulp cells and the formation of pulp dentin complex. This study provides a potential application of MBG that repairing and regenerating for dental tissue injurie. However, several points need to be clarified.
1. The information of reagents, materials and instruments that mentioned in this manuscript should be supplied, such as manufacturer, code and country.
2. The identification for hPDCs should be provide.
3. Table 2, the mean and standard deviation of ion concentration of MBG extract should be considered,
4. The author mentioned that MBG has a large specific surface area, therefore the specific surface area and mesoporous size should be measured.
5. The Figure 3 cannot support the author’s finding, like mineralized matrix with dentin tubule structure, and the residual MBG particles can’t be observed in the Figure 3. The author should provide more evidence to support the correctness of results.

Experimental design

no comment

Validity of the findings

no comment

Additional comments

no comment

---

## Round 0.2 · accepted · Accept

Dear Dr. Zhu,

The manuscript was significantly improved and could be accepted for publication, congratulations!

Reviewer 1 ·

Basic reporting

The authors have well addressed my comments and questions. The paper can be accepted for publication.

Experimental design

The authors have well addressed my comments and questions. The paper can be accepted for publication.

Validity of the findings

The authors have well addressed my comments and questions. The paper can be accepted for publication.

Reviewer 2 ·

Basic reporting

The revised manuscript has addressed most the comments and suggestions by the reviewers, thus, the reviewer recommends publication of current manuscript.

Experimental design

No comment

Validity of the findings

No comment